DOI: 10.1038/s41467-017-01202-2 | **OPEN**

# Hydrogen substituted graphdiyne as carbon-rich flexible electrode for lithium and sodium ion batteries

Jianjiang He[1,2], Ning Wang[1], Zili Cui[1], Huiping Du[1,2], Lin Fu[1,2], Changshui Huang [1], Ze Yang[1], Xiangyan Shen[1,2], Yuanping Yi [3], Zeyi Tu[3] & Yuliang Li[3]

Organic electrodes are potential alternatives to current inorganic electrode materials for lithium ion and sodium ion batteries powering portable and wearable electronics, in terms of their mechanical flexibility, function tunability and low cost. However, the low capacity, poor rate performance and rapid capacity degradation impede their practical application. Here, we concentrate on the molecular design for improved conductivity and capacity, and favorable bulk ion transport. Through an in situ cross-coupling reaction of triethynylbenzene on copper foil, the carbon-rich frame hydrogen substituted graphdiyne film is fabricated. The organic film can act as free-standing flexible electrode for both lithium ion and sodium ion batteries, and large reversible capacities of 1050 mAh g$^{-1}$ for lithium ion batteries and 650 mAh g$^{-1}$ for sodium ion batteries are achieved. The electrode also shows a superior rate and cycle performances owing to the extended π-conjugated system, and the hierarchical pore bulk with large surface area.

[1] Qingdao Institute of Bioenergy and Bioprocess Technology, Chinese Academy of Sciences, No. 189 Songling Road, Qingdao 266101, China. [2] University of Chinese Academy of Sciences, No. 19A Yuquan Road, Beijing 100049, China. [3] Beijing National Laboratory for Molecular Sciences (BNLMS), CAS Key Laboratory of Organic Solids, Institute of Chemistry, Chinese Academy of Sciences, Beijing 100190, China. Jianjiang He and Ning Wang contributed equally to this work. Correspondence and requests for materials should be addressed to C.H. (email: huangcs@qibebt.ac.cn) or to Y.L. (email: ylli@iccas.ac.cn)

I n recent years, the portable and wearable devices have undergone important technological developments. High specific capacity and good conductivity pliable electrode with flexibility and bendable current collectors are crucial features of lithium ion batteries (LIBs) and sodium ion batteries (SIBs)[1–3]. The organic electrode materials have attracted interest from scientists since the first report by Williams and co-workers in 1969 due to their design versatility, flexibility, low cost, and environmentally friendly properties[4–7]. However, the major issues facing conducting polymer based organic materials are the low energy density and stability. Furthermore, other organic materials like organodisulfides, organic free radicals, and carbonyls suffer from the intrinsic low electrical conductivity and high solubility in traditional non-aqueous electrolytes[8–10]. Cross-linked carbon-rich or all-carbon frame could show exceptional thermal and chemical stability, good conductivity, high degree of strength, and unusual mechanical properties such as strong shear deformation modes. These make them to be promising candidates for LIB and SIB electrodes.

Nowadays, the most valuable all-carbon material is graphite, which has been commercialized as anode in LIBs for a long time. Its good electrochemical performance benefits from the acceptable capacity of $372\ \mathrm{mAh\ g^{-1}}$ caused by the insertion of lithium ions in interlayer and the stable conversion structure of $LiC_6$ originate from weak van der Waals interaction. However, Li ion diffusion parallel to the plane of graphite is limited by the steric hindrance, and diffusion perpendicular to the basal plane is hindered by the aromatic carbon rings, thus leading to a low power density[11]. Although varies of all-carbon materials were fabricated to improve the electrochemical performance, including graphene, hard carbon, carbon nanotubes and so on, the molecular structure problem remained unsolved[12, 13]. While recently, our group is pioneer in developing new carbon allotropes graphdiyne as a high-capacity electrode for LIBs and SIBs. It gives new insight into the layered material electrodes[14–19]. This novel material has the merits of good electronic conductivity like graphite and pore structure and synthesizable process of polymers. The butadiyne linkages between the repeating hexatomic benzene matrixes grant graphdiyne uniformly distributed pores, and low Li ion diffusion energy barrier. Based on various study of applying graphdiyne as electrode for lithium ion batteries, in order to improve the capacity in lithium storage for carbon materials, there are still two reasonable aspects can be considered. For one thing, the larger pore size in the molecular structure is in favor of ion diffusion in bulk materials which may become tardy due to the AB stack of graphdiyne[20]. For another, small ratio of H atoms is benefit for a larger capacity as it is reported that Li atoms can bind in the vicinity of H atoms in these hydrogen-containing carbons[21, 22].

Herein, we synthesize a carbon-rich framework which is named as hydrogen substituted graphdiyne (HsGDY). It is applied as a flexible electrode for LIBs and SIBs. HsGDY is an extended π-conjugated carbon skeleton comprised of butadiyne linkages and benzene rings, while aromatic hydrogen (Ar-H) group of benzene ring is introduced to HsGDY to provide more active binding sites for Li/Na storage. This free-standing HsGDY film electrode can achieve a highly improved reversible capacity of $1050\ \mathrm{mAh\ g^{-1}}$ for LIBs and $650\ \mathrm{mAh\ g^{-1}}$ for SIBs at a current density of $100\ \mathrm{mA\ g^{-1}}$. The sodium storage performance and stability of HsGDY are among the best values reported for flexible batteries. Remarkably, this hierarchical porous structure with high π-conjugation of HsGDY leads to high rate performance and can reach $570\ \mathrm{mAh\ g^{-1}}$ for LIBs and $220\ \mathrm{mAh\ g^{-1}}$ for SIBs at the rate as high as $5\ \mathrm{A\ g^{-1}}$.

## Results

**Chemical structure of HsGDY.** HsGDY is synthesized through an in situ cross-coupling reaction of triethynylbenzene on copper foil in pyridine as a large-area free-standing film (Fig. 1a). The structure and purity of the triethynylbenzene monomer can be confirmed by the $^1H$ NMR and $^{13}C$ NMR (Supplementary Figs. 1 and 2). The photo of HsGDY film is shown in Fig. 1b. The film is faint yellow and almost transparent with length of 4 cm and width of 3 cm. HsGDY is a carbon-rich polymer, with unit composed of 42-C hexagons by connecting six benzene rings through butadiyne linkages ($-C\equiv C-C\equiv C-$). Different from graphene and graphdiyne, it has lower atom density, larger pores and H group in the pores, which would lead to higher Li storage capacity and excellent Li mobility. Structure of HsGDY possesses a large π-conjugation system which would satisfy the need to achieve a good conductivity for batteries. Since HsGDY was synthesized based on the structure of γ-GDY, but with different initial monomer, the synthesized material HsGDY would like a new carbon-rich material rather than a new phase of GDY. In consideration of the important role of phase information in Li storage capacity, HsGDY may be classed as hydrogen substituted γ-GDY rather than α-GDY or β-GDY[15, 23, 24].

It is observed in Fig. 1c that four carbon species and four functional groups exist in the structure of HsGDY. The solid-state NMR result unequivocally indicates the chemical structure of HsGDY. Four kinds of carbon mainly exist in the HsGDY framework (Fig. 1d). The peaks at 123.1 and 135.7 p.p.m. correspond to aromatic C–C and C–H sites. The peaks at 75.5 and 81.1 p.p.m. can be ascribed to C(sp)-C(sp) and C(sp)-C(sp²) sites. Figure 1e shows the XRD result of the as-synthesized sample. The peaks at 21.2° for HsGDY can be corresponded to the interlayer spacing of 4.19 Å. Although the presence of HsGDY framework is confirmed, its crystallinity is poor due to conformational fluctuation of HsGDY at mesoscopic scales. The carbon species were also characterized by X-ray photoelectron spectroscopy (XPS) in Supplementary Fig. 3. In detail, the C1s peaks of HsGDY in Fig. 1f can be deconvoluted into four subpeaks of C–C (sp²) at 284.7 eV, C–C (sp) at 285.4 eV, C–O at 287.5 eV and C=O at 288.4 eV, respectively[15, 25]. XPS data shows that the synthetic samples have both sp² and sp hybrid carbon, and the area ratio of the two is close to 1:1, consistent with the structure shown in Fig. 1c. The XPS through the peak fitting of C1s (Fig. 1f) and Fourier transform infrared spectroscopy (FT-IR) measurement (Fig. 1g) show the existence of C–O and C=O bonds, but only weak peaks could be observed in solid-state NMR around 67.5 and 166.0 ppm (Fig. 1d). The results indicated the existence of small amount of C–O and C=O bonds on the surface of HsGDY samples. The origin of the C–O and C=O bonds might be ascribed to the chemical adsorption of oxygen on the surface of HsGDY and the reaction between oxygen and some exposed terminal acetylenic bond, which was also observed in other carbon materials[15, 25].

The Raman spectrum in Fig. 1g exhibits three main peaks. A G-band at $1584\ \mathrm{cm^{-1}}$ suggests the samples possess abundant aromatic rings, and a D-band at $1356\ \mathrm{cm^{-1}}$ is corresponding to defects and edges[26]. A weak peak at $2217\ \mathrm{cm^{-1}}$ is ascribed to acetylenic bond[27]. Figure 1h shows the FT-IR spectrum of as-grown HsGDY. The peaks in $1500–1650\ \mathrm{cm^{-1}}$ are assigned to the skeletal vibrations of aromatic ring. The peak at $2200\ \mathrm{cm^{-1}}$ is the typical $C\equiv C$ stretching vibration. The stretching and bending vibration of aromatic C–H also can be observed at 3053 and 883 $\mathrm{cm^{-1}}$, respectively. The peaks at $1000–1500\ \mathrm{cm^{-1}}$ may be contributed to the asymmetric stretching mode of C-C. All of these characterization results indicate the formation of the carbon-rich structure. In contrast to the FT-IR spectrum of the triethynylbenzene monomer (Supplementary Fig. 4), it is clearly

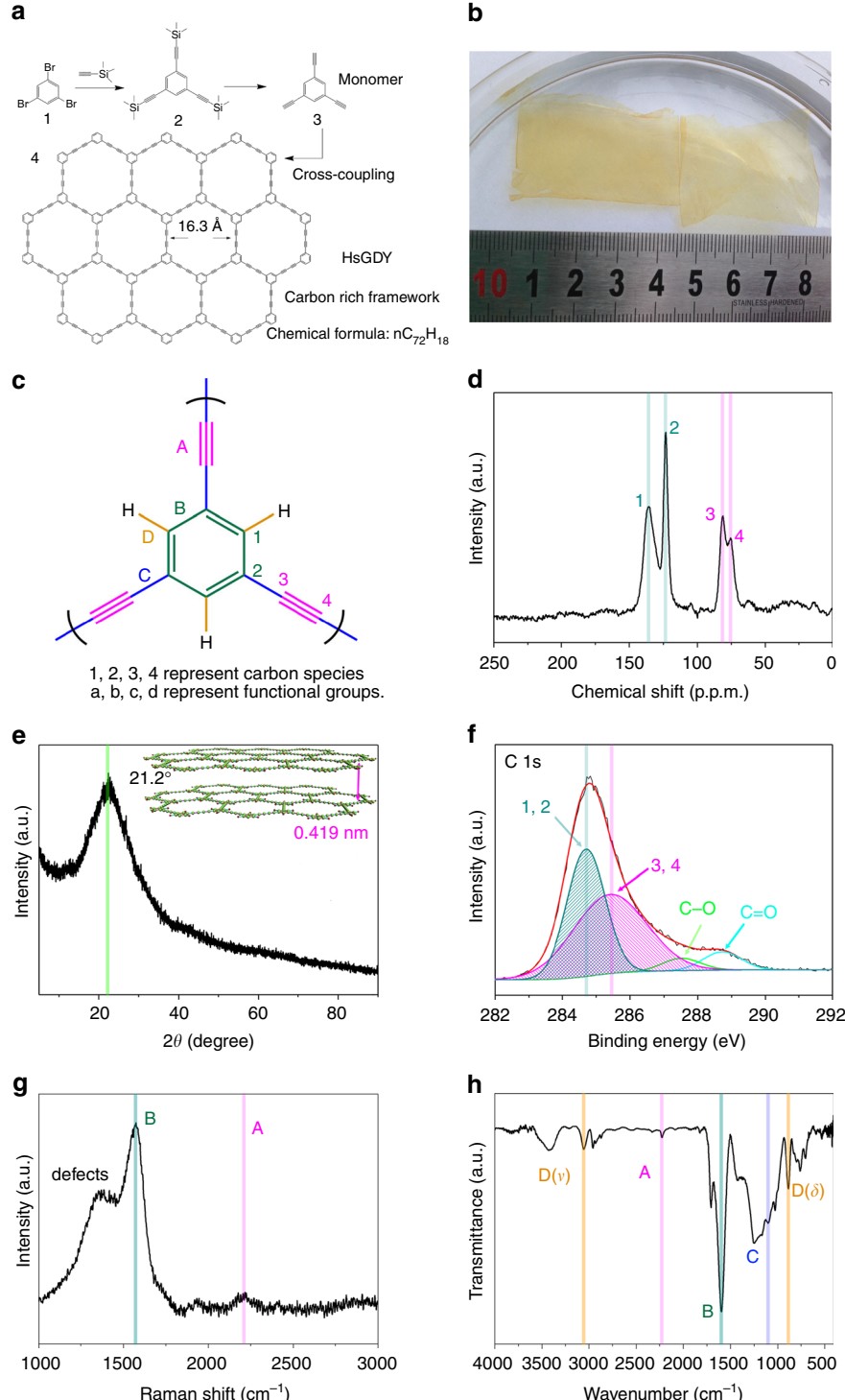

**Fig. 1** The structure and appearance of HsGDY. **a** Schematic diagram of the synthesis of the HsGDY, (1) Tribromobenzene, (2) Tris[(trimethylsilyl)ethynyl]benzene, (3) Triethynylbenzene, (4) carbon-rich framework HsGDY. **b** The photograph of free-standing HsGDY films. **c** Carbon species and functional groups in HsGDY. **d** $^{13}$C solid-state NMR spectrum of HsGDY. **e** XRD patterns of HsGDY. **f** XPS spectrum of HsGDY. **g** Raman spectrum of HsGDY and **h** FT-IR spectrum of HsGDY

observed the decreasing intensity of the acetylenic C-H vibration at 3279 cm$^{-1}$ which is corresponding to the cross-coupling reaction of triethynylbenzene. The extended graphitic C≡C stretching vibration in G-band also can be found in Raman spectrum of HsGDY (Supplementary Fig. 5). The UV–vis absorption spectrum was used to characterize the electronic conductivity of HsGDY (Supplementary Fig. 6). The energy gap

of the thin film between HOMO (highest occupied molecular orbit) and LUMO (lowest unoccupied molecular orbit) was measured to be 0.75 eV by the following relation $\alpha \propto (h\nu\text{-}E_{g})^{1/2}/h\nu$[28]. The energy gap is smaller than many other organic electrodes[7, 9, 29]. The low carrier barrier can be overcome under the experiment conditions. Furthermore, the current–voltage (I–V) curve at a bias voltage from −3 to 3 V was also measured in

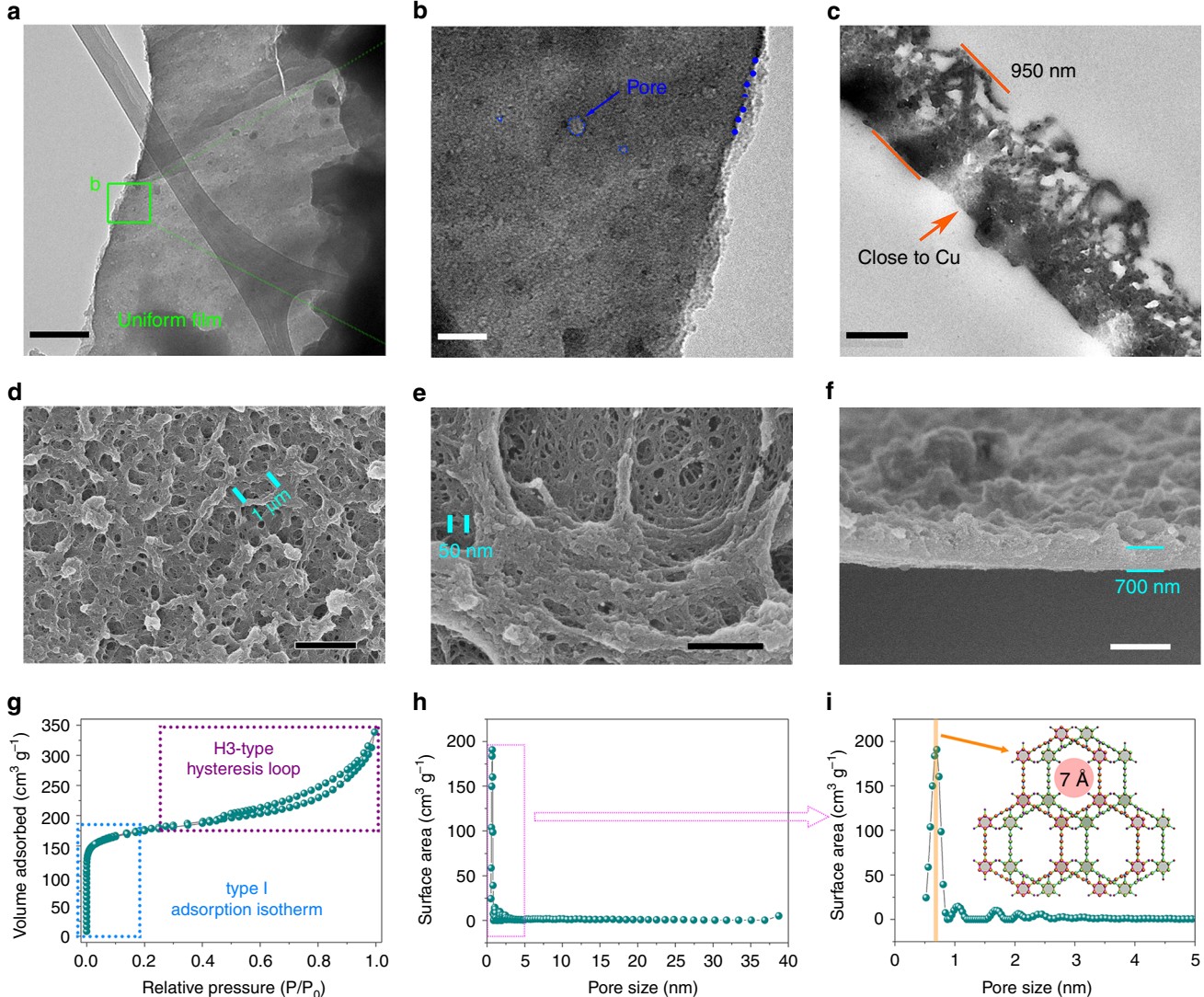

**Fig. 2** The morphology of HsGDY thin film. **a**, **b** The TEM images of HsGDY, and **c** the cross-section TEM images of HsGDY. **d**, **e** The SEM images of HsGDY and **f** the cross-section SEM images of HsGDY. **g** Nitrogen adsorption-desorption isotherm and **h**, **i** the corresponding DFT incremental pore size distribution curve for HsGDY. Scale bar, 200 nm, **a**; 50 nm, **b**; 500 nm, **c**, **e**; 2 μm, **d**, **f**

Supplementary Fig. 7. It can be seen that the I-V curve of HsGDY is approximately linear, which exhibits semiconductor behavior and the slope of the lines are fitting as 0.0146. The conductivity of HsGDY is calculated to be $1.02 \times 10^{-3}$ S m$^{-1}$. The small energy gap and high electronic conductivity of HsGDY imply that the extended π-conjugated system is set up through butadiyne linkages and benzene rings.

**Morphology of HsGDY**. The stacked sheets can be clearly observed in the TEM images of HsGDY. The exfoliation of thick layers is difficult due to the strong interaction. A uniform film is shown in Fig. 2a suggesting the regular structure of HsGDY. The higher resolution image is observed from Fig. 2b. The boundary of the HsGDY layer is clearly visible as marked with the blue dashes. Moreover, a large number of micropores can be observed. The pore diameter is in the range from 2 to 15 nm which correspond to about 1 to 8 monomers. The existence of these pores ascribes to the elimination of oligomeric monomers. These pores not only are beneficial for a large specific surface

area, but also can effectively reduce lithium diffusion barrier heights through the basal plane and thus enhance the power density. TEM image in Fig. 2c is obtained from cross-section of the sample. The thickness of the free-standing thin film is about 950 nm which is a little larger than the observed value in cross-section SEM images. This may be due to the insertion of embedding medium. An interesting phenomenon that the morphology of thin film at the side close to the copper is flat while that at the side away from the copper is irregular can be found in cross-section TEM image of HsGDY, which is ascribed to the catalytic activity of cooper foil. It was reported that the copper foil not only played the role as a substrate but also acted as a catalyst in the synthetic process[15, 25]. Hence, HsGDY is growing and overspread on the surface of copper foil along with the reaction proceeding to form the film. However, the subsequent catalytic reaction would be inhibited by the formed HsGDY film.

The surface morphology of HsGDY is observed in SEM images (Fig. 2d). As shown in Fig. 2d, the sample possesses abundant of pores ranging from macroporous to mesoporous. The pore size of macropores and mesopores is around several micrometers and

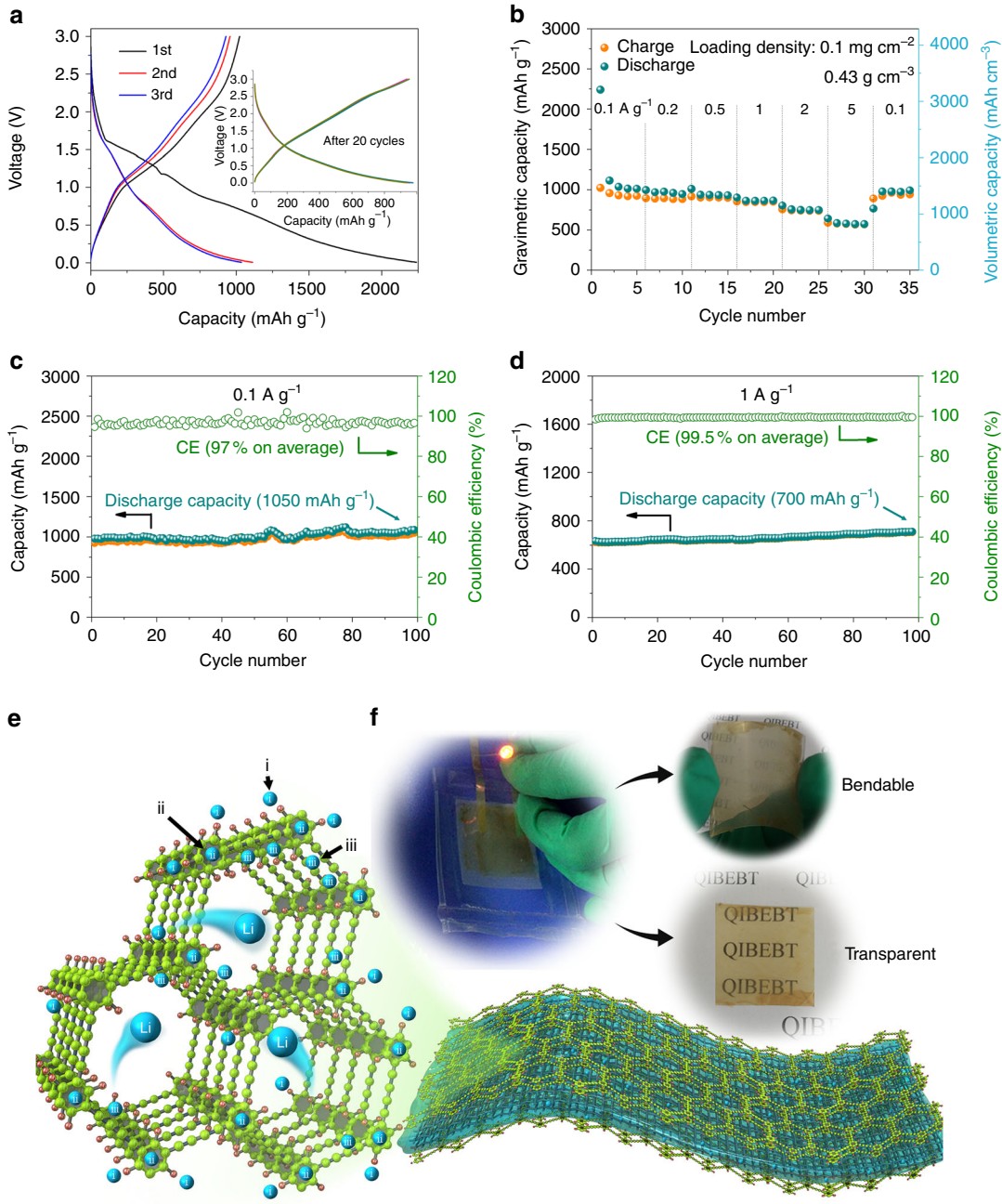

**Fig. 3** The electrochemical performance of HsGDY electrodes in Li metal half-cell format. **a** The charge–discharge profiles of the HsGDY based electrodes at the current density of 0.1 A g$^{-1}$. **b** The rate performance of the flexible electrode for LIBs. The cycle performance of flexible electrode at the current density of **c** 0.1 A g$^{-1}$ and **d** 1 A g$^{-1}$. **e** The mechanism of Li storage. **f** A bendable transparent LIB is made up of HsGDY. (All the potential is vs. Li$^+$/Li)

tens of nanometers. It is also observed that the HsGDY sample connected together to form 3D porous film. Specifically, as depicted in Fig. 2e, a lot of mesopores about 50 nm in diameter can be observed. Sufficient and fast ionic diffusion could be achieved by this introduction of meso/macropores second-order structure[30]. The cross-section SEM image in Fig. 2f reveals that the thickness of the sample is approximately 700 nm which is consistent with the observation in TEM image. Meanwhile, the same phenomenon as TEM image is worth noting that the one side of the thin films close to the copper foil is flat while the other is unevenness. The loading density of the film weighs about 0.10–0.11 mg cm$^{-2}$ with the thickness of 700 nm. The morphology of HsGDY film is similar with each other by different

synthetic batches, which demonstrates the good reproducibility of carbon-rich framework (Supplementary Fig. 8). Moreover, the thickness and areal density of the HsGDY films can be tuned by controlling the initial amount of monomer. As shown in Supplementary Fig. 9f, the thickness of the film is 2.5 μm with the initial monomer amount of 0.266 mmol (twice the weight of the original HsGDY film with 700 nm thickness). The loading density of electrode based on this HsGDY film increases to 0.18–0.20 mg cm$^{-2}$. In the meantime, the pore size become larger and the film become more porous (Supplementary Fig. 9d and e). This phenomenon can be ascribed to the catalytic reaction which is discussed earlier in the article. However, the thickness of the film as grown is changed slightly (only 2.9 μm) when the

| Table 1 Comprehensive overview of carbon-based anode materials in LIBs | | | | |
|---|---|---|---|---|
| **Electrode** | **Reversible capacity (mAh g$^{-1}$)** | **Discharge rate (mA g$^{-1}$ or C)** | **Loading density** | **Reference** |
| HsGDY | 1050 | 100 | 0.1–0.11 mg cm$^{-2}$(1.43 g cm$^{-3}$) | This work |
| HsGDY | 700 | 1000 | 0.1–0.11 mg cm$^{-2}$ (1.43 g cm$^{-3}$) | This work |
| NG | 360 | 15 | — | ref. [44] |
| CNT | 200 | 400 | 0.025 mg cm$^{-2}$ | ref. [45] |
| Graphene | 460 | 1 C | — | ref. [46] |
| N-graphene | 872 | 50 | 20.8–24.2 mg cm$^{-3}$ | ref. [47] |
| B-graphene | 700 | 500 | 20.8–24.2 mg cm$^{-3}$ | ref. [47] |
| N-CNF | 1280 | 100 | — | ref. [48] |
| N-CNT | 516 | 200 | 0.796 mg cm$^{-2}$ | ref. [49] |
| PAA | 995 (1st cycle) | 100 | — | ref. [50] |
| CLP | 619 | 100 | — | ref. [51] |
| MCOF | 74 | 2.4 C | 0.69 mg cm$^{-2}$ | ref. [52] |

B-graphene B-doped graphene, CLP conjugated ladder polymers, CNT carbon nanotube, MCOF mesoporous covalent organic framework, N-CNF N-doped porous carbon nanofiber, N-CNT N-doped core-sheath carbon nanotube, NG natural graphite, N-graphene N-doped graphene, PAA polyazaacene analog

monomer amount is threefold the weight of the monomer preparing HsGDY film with 700 nm (Supplementary Fig. 9i). The loading density of electrode based on this HsGDY film is about 0.25–0.26 mg cm$^{-2}$. Since the copper works as both the support and catalyst, the thicker the film grows the more difficult the catalytic reaction occurs. It also can be seen from Supplementary Fig. 9g and h that the porous film collapse and the pore become smaller than the other two films. All the volume densities (tap density) of the HsGDY film with thickness of 2.5 and 2.9 μm are smaller than that of HsGDY film with 700 nm.

To gain a better understanding of the pore structure for the samples, nitrogen adsorption-desorption studies were performed. It can be seen from Fig. 2g that the adsorption quantity of N$_2$ was sharply increased at very low P/P$_0$ due to a mass of single layer adsorption of N$_2$ in the micropores. The continued adsorption of N$_2$ and H3-type hysteresis loop at relatively higher pressure indicating the multilayer adsorption which can be ascribed to the existence of mesopores and macropores. The Brunauer-Emmett-Teller (BET) surface area for the sample is 667.9 m$^2$ g$^{-1}$. The pore size distributions of the samples calculated from the corresponding nitrogen adsorption-desorption isotherms using the density functional theory (DFT) method are shown in Figs. 2h and i. The main pore size distribution for HsGDY is about 0.7 nm (Fig. 2i) indicating a highly ordered porous structure in framework. It is reported that the AB stack is always present in the 2D materials because of the energetically favorable[20, 23]. Moreover, it is reported that there is a strong interaction between C–H and C≡C triple bond[31]. As a result of the theoretical pore size of single layer (1.63 nm) in Fig. 1a, this experimental pore size of 7 Å is in accordance with the AB stack of HsGDY layer (Fig. 2i). The distribution pore size at 40 nm also can be observed (Fig. 2g). These hierarchical pores in HsGDY could be guaranteed to achieve high electrochemical performance for LIBs and SIBs.

**HsGDY applied for LIBs**. The electrochemical performances of the free-standing flexible electrode was evaluated using half cells with lithium metal as reference electrode in the potential range of 0.005–3 V vs. Li$^+$/Li (Fig. 3). The charge–discharge profiles of the initial three cycles are shown in Fig. 3a at the current density of 0.1 A g$^{-1}$. The largest discharge capacity is obtained under 1 V, which contains irreversible capacity in the first cycle. The Coulombic efficiency (CE) is measured to be 45.7%. The high irreversible capacity loss can be ascribed to the formation of extensive solid electrolyte interface (SEI) layers on high interface area in the first discharge process. The charge–discharge profiles of the electrode are identical after 20 cycles suggesting the good

electrochemical stability of HsGDY (inset of Fig. 3a). Figure 3b shows the excellent rate performance of the flexible electrode from 0.1 to 5 A g$^{-1}$. The reversible gravimetric capacity and volumetric capacity of the HsGDY are 1012 mAh g$^{-1}$ and 1447 mAh cm$^{-3}$ at the current density of 0.1 A g$^{-1}$, and can even achieve 570 mAh g$^{-1}$ and 815 mAh cm$^{-3}$ while the current density increased to 5 A g$^{-1}$. These experimental capacities are much greater than the theoretical gravimetric and volumetric capacities of 372 mAh g$^{-1}$/818 mAh cm$^{-3}$ and 744 mAh g$^{-1}$ for graphite and γ-graphdiyne suggesting that HsGDY can serve as a promising high-capacity lithium ion battery anode. In addition, the capacity can be fully recovered after cycled at various rate. The reversible capacity can reach 1050 and 700 mAh g$^{-1}$ after 100 cycles at the current density of 0.1 and 1 A g$^{-1}$ (Figs. 3c, d). The CE of the flexible electrode is ~97% in average at the current density of 0.1 A g$^{-1}$ and ~99.5% in average at the current density of 1 A g$^{-1}$. The electrochemical performance of HsGDY is higher than that of intrinsic carbon allotropes, N-doped Graphene, organic molecules, and comparable to that of N-doped porous carbon nanofiber in LIBs (Table 1). The superior electrochemical performance of pore structure also plays important roles in the performance of HsGDY. In this case, HsGDY can be attributed to the high surface area and conductivity. The hierarchical abundant micropores and mesopores endow the electrode a high specific surface area, giving rise to a large reversible capacity and high energy density. Furthermore, the linked mesopores and macropores shorten the diffusion pathway and facilitate the ion transport, which bring about high rate performance and power density[32]. The reproducibility of the electrochemical performance for HsGDY electrodes in LIBs is investigated in Supplementary Fig. 11a–c. It is observed that the rate performance and cycle performance of the different HsGDY film is almost same. The similar phenomenon is found in SIBs (Supplementary Fig. 11d–f). Meanwhile, the electrochemical performance of HsGDY films with different density mentioned above was also measured. The reversible capacity and rate performance of HsGDY film in LIBs slightly reduce with the increasing of areal density (Supplementary Fig. 12). Those can be ascribed to the low tap density and poor quality of the thick films. On the other hand, high loading density (2.1 mg cm$^{-2}$) of HsGDY electrode is achieved by coating HsGDY powder on copper foil with 10% of Super P as conductive agent and 10% of PVDF as binder. The powder was obtained by grinding the HsGDY film. The electrodes exhibit a comparative capacity with that of free-standing HsGDY film (Supplementary Fig. 14a). But this kind of HsGDY based electrode was not free-standing, bendable and transparent, comparing with HsGDY film based electrode.

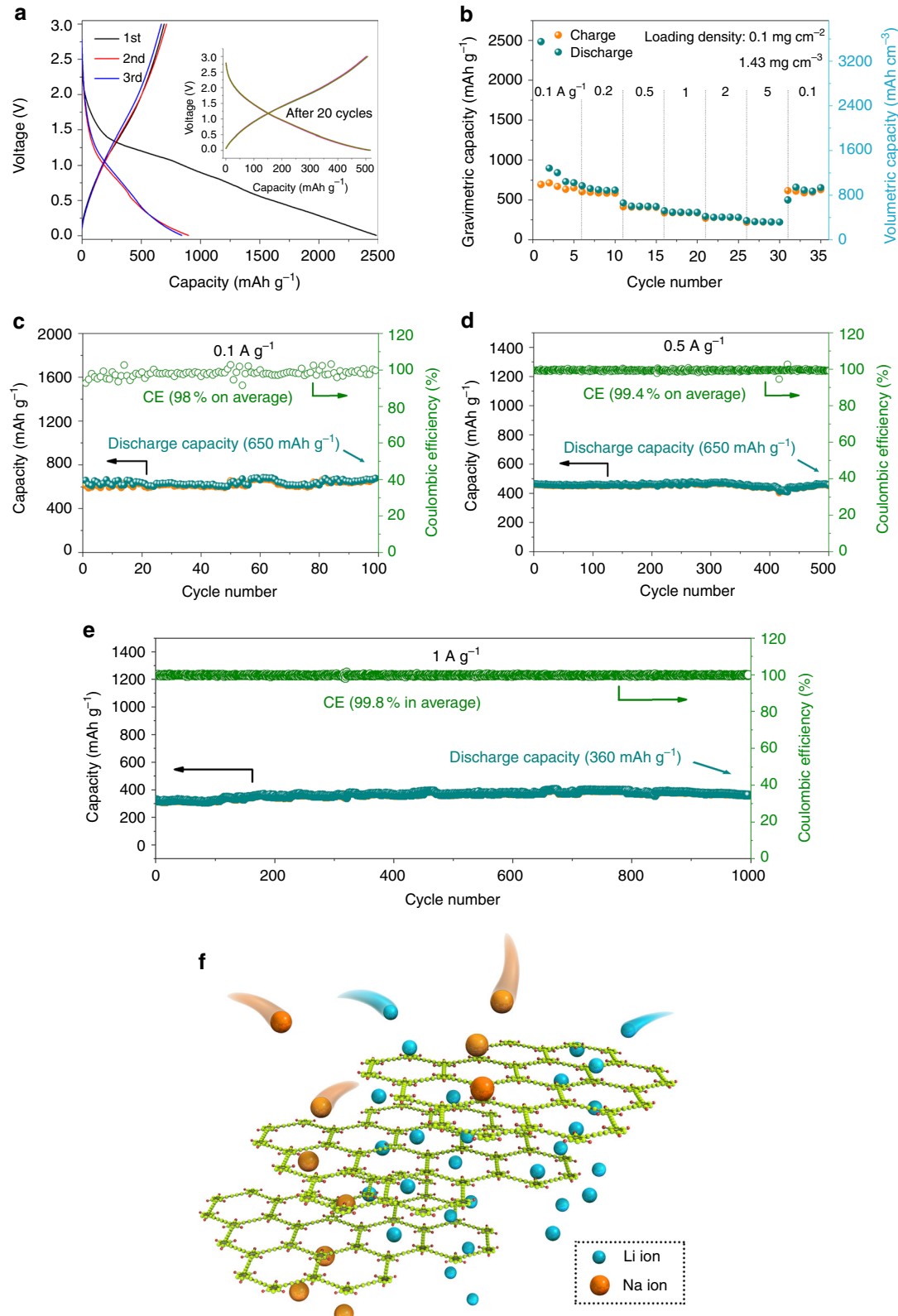

**Fig. 4** The electrochemical performance of HsGDY electrodes in Na metal half-cell format. **a** The charge–discharge profiles of the HsGDY based electrodes at the current density of 0.1 A g⁻¹. **b** The rate performance of the flexible electrode for SIBs. The cycle performance of flexible electrode at the current density of **c** 0.1 A g⁻¹, **d** 0.5 A g⁻¹ and **e** 1 A g⁻¹. **f** the diffusion path of Li ions and Na ions in carbon-rich framework. (All the potential is vs. Na⁺/Na)

**Table 2 Comprehensive overview of carbon-based anode materials in SIBs**

| Electrode | Reversible capacity (mAh g⁻¹) | Discharge rate (mA g⁻¹ or C) | Cyclic stability (cycles) | Loading density | Reference |
|---|---|---|---|---|---|
| HsGDY | 650 | 100 | 100 | 0.1–0.11 mg cm$^{-2}$ (1.43 g cm$^{-3}$) | This work |
| HsGDY | 460 | 500 | 500 | 0.1–0.11 mg cm$^{-2}$ (1.43 g cm$^{-3}$) | This work |
| HsGDY | 360 | 1000 | 1000 | 0.1–0.11 mg cm$^{-2}$ (1.43 g cm$^{-3}$) | This work |
| HC | 326 | C/10 | 100 | — | ref. [39] |
| HCNW | 251 | 50 | 400 | — | ref. [38] |
| NG | 100 | 500 | 2500 | 4.3 mg cm$^{-2}$ (0.96 g cm$^{-3}$) | ref. [37] |
| N-GF | 594 | 500 | 150 | 1 mg cm$^{-2}$ (0.22 g cm$^{-3}$) | ref. [41] |
| PCG | 400 | 50 | 100 | — | ref. [30] |
| PCNF | 240 | 100 | 100 | — | ref. [53] |
| N-CNF | 377 | 100 | 100 | 0.64 mg cm$^{-2}$ (0.14 g cm$^{-3}$) | ref. [54] |
| NaTP | 295 | C/10 | 100 | — | ref. [55] |
| SSDC | 112 | 1000 | 400 | 1 mg cm$^{-2}$ | ref. [29] |

HC hard carbon, HCNW hollow carbon nanowires, NaTP sodium terephthalate, N-CNF N-doped carbon nanofiber, NG natural graphite, N-GF N-doped graphene foams, PCG porous carbon/graphene, PCNF porous carbon nanofiber, SSDC sodium 4,4′-stilbene-dicarboxylate

**Li storage mechanism in HsGDY.** Three main storage modes are proposed based on previous reported works. As marked in Fig. 3e, (i) Li atoms can be bound near the H atoms in this hydrocarbon. This always causes changes in the defined atom positions of the C and H atoms, and the bonding distortion would be an activated process, which results in hysteresis[21]. The substantial work that has been done on ternary graphite-alkali-hydrogen materials showed that the charge transfer from alkalis to hydrogen in carbons is expected. Although the Li storage mechanism for various hydrogen-terminated carbon materials is still unclear, many references have reported that lithium atom could bind to H atom at about 0.7–1.5 V which is in accordance with the inconspicuous plateau in cyclic voltammogram (CV) and charge–discharge curves (Fig. 3a and Supplementary Fig. 10a)[33]. (ii) Li atoms can be adsorbed above the center of 6-C hexagon[21, 34]. This Li storage mode exists in most graphite electrode. (iii) Li atoms can be located at the vicinity of butadiyne in the hexagonal pore[34, 35]. It is also reported that high specific capacity is obtained by storage of lithium in micropores[36]. In this work, HsGDY delivers considerable capacity at the plateau of near 0 V observed in CV and charge–discharge curves (Fig. 3a and Supplementary Fig. 10a). This can be attributed to the insertion of Li above the benzene ring (mode ii) and the storage of Li in the micropores with main size distribution of 0.7 nm (mode iii). The inconspicuous peaks in the CV of the sample imply that the capacity is mainly dominated by faradaic pseudocapacitance rather than redox reaction because of the high specific surface area.

In the meantime, the theoretical calculations of Li storage were performed using a first-principles method based on density functional theory. The definition of binding energy and the binding energy of single Li atom at different adsorption sites on selected repeating unit can be found in Supplementary Fig. 15. More details of theoretical calculations and results are provided in Supplementary Fig. 15–17. Briefly, as shown in Supplementary Fig. 16, the stabilities of $Li_{28}$-$C_{24}H_6$ complex was examined by their binding energies on single layer HsGDY. It can be observed that the Li atoms are stored at the nearby of H atoms, benzene ring and butadiyne in calculation, which are in accordance with that in reference[21, 33–36]. Moreover, the Li storage capacity is calculated to be 2553 mAh g⁻¹ in which the adsorption in both sides of HsGDY layer is required. That the calculated capacity is larger than experimental result can be ascribed to the omission of steric hindrance between the layers in AB-stacking multilayer HsGDY.

The calculated Li storage capacity of HsGDY is lower than that of α-graphdiyne and higher than that of γ-graphdiyne. This is because α-graphdiyne is mainly comprised by carbyne which always shows much higher capacity than benzene ring. Hence, the theoretical capacity of α-graphdiyne may be highest among 2D layers of sp-sp$^2$ hybrid carbon networks. On the other hand, hydrogen is introduced into γ-graphdiyne to stabilize the structure, enlarge the pore size and provide more active binding sites. Therefore, as far as we know, HsGDY delivers the highest experimental capacity among the synthesized sp-sp$^2$ hybrid carbon networks.

In the article, the HsGDY frame represents a matrix, rather than a current collector, to interconnect numerous redox sites for the storing of ions by a self-exchange mechanism, leading to the storage and transport of charge in a homogeneous solid. Moreover, the carbon-rich framework endows HsGDY with excellent mechanical property which can act as a bendable transparent electrode in LIB as shown in Fig. 3f.

**HsGDY applied for SIBs.** Figure 4 shows the electrochemical performance of the as-synthesized sample as anode materials for SIBs. The charge–discharge profiles of the flexible electrode at the current density of 0.1 A g⁻¹ are presented in Fig. 4a. It can be observed the electrochemical process of the sample in SIBs is similar to that in LIBs. The largest discharge capacity is obtained under 1 V and the CE is measured to be 27.9%. The charge–discharge profiles of the electrode are identical after 20 cycles, suggesting that HsGDY is stable in SIBs (inset of Fig. 4a). The CV of the electrode in SIBs is similar to that in LIBs indicating the same electrochemical process (Supplementary Fig. 10b). The rate performance of the flexible electrode is shown in Fig. 4b. The reversible capacity is 220 mAh g⁻¹ for the electrode at the current density of 5 A g⁻¹, which benefit by the good conductivity. As shown in Figs. 4c, d, e, long-cycle performance of the flexible electrode was investigated. The reversible capacity can reach 650 mAh g⁻¹ at the current density of 0.1 A g⁻¹ after 100 cycles. Meanwhile, the CE for the flexible electrode is 98% in every cycle. When the current density increases to 0.5 and 1 A g⁻¹, the reversible capacity still maintains at 460 mAh g⁻¹ after 500 cycles and 360 mAh g⁻¹ after 1000 cycles. They are much higher than hard carbon, hollow carbon nanowires, expanded graphite, organic compound and so on (Table 2)[37–40]. Although the N-doped graphene foams deliver a higher reversible capacity, it suffers quickly degradation of capacity after 150 cycles[41]. Furthermore, the CE of HsGDY electrodes can achieve 99.4 and 99.8% at the current density of 0.5 and 1 A g⁻¹, respectively

(Figs. 4d, e). The application of the HsGDY films with other different density for SIBs got similar results (Supplementary Fig. 13 and Supplementary Fig. 14b). The excellent electrochemical performance of the carbon-rich framework based electrode in SIBs and LIBs indicates that the pore structure and specific surface area play important roles in SIBs and LIBs (Fig. 4f). In general, most electrode materials of LIBs do not have sufficiently big interstitial space within their bulk materials to host and transport Na ions in terms of larger Na ion than Li ion[38]. In our case, the designed structure provides favorable path and sites to satisfy the diffusion and insertion/extraction of large diameter ions. To further understand the Na storage in HsGDY electrodes, rational configuration of HsGDY as shown in Supplementary Fig. 17 was selected for the theoretical calculation of Na storage[18, 42, 43]. It can be found that a higher binding energy was got in the optimized $Na_{22}-C_{24}H_6$ configuration in comparison with $Li_{28}-C_{24}H_6$ configuration. It can be attributed to that strong repulsion among Na atoms and the large diameter make the substantial storage of Na in hexagonal pore difficult. This phenomenon is also observed in other carbon materials with many micropores[22].

## Discussion

Supplementary Fig. 18a and b show the morphologies of the electrodes for LIBs and SIBs after cycles. It can be seen that a thick SEI layer was formed on the surface of the electrode, resulting in a low initial CE. However, this thick SEI layer can effectively prevent HsGDY from the side reaction with electrolyte and thus keep the reversible capacity stable. In Supplementary Fig. 19, the peaks in Raman spectrum of the sample after cycles is similar with that of the fresh sample. The result implies the destruction of the structure is negligible during cycles.

Electrochemical impedance spectroscopy (EIS) was further carried out to understand the interfacial charge transfer and Li ions diffusion process in the electrode. Supplementary Fig. 18c shows Nyquist plots of the sample for LIBs after 3 and 100 cycles. And the fitting kinetic parameters of the electrodes are listed in Supplementary Table 1. It can be observed that there is a huge reduction in SEI resistance ($R_s$) and charge transfer resistance ($R_{ct}$) of the electrode after cycles. This phenomenon can be attributed to the stable SEI layer formed after the initial cycle, which can effectively reduce the interface resistance and thus stable the battery system.

To gain a further understanding of the excellent cycle performance of HsGDY, the CVs and EIS for SIBs after different cycles are conducted in Supplementary Fig. 20 and 21. The very similar curves demonstrate excellent stability of interface between free-standing HsGDY and electrolyte due to the formation of thick SEI films.

In summary, HsGDY was fabricated through a cross-coupling reaction. This carbon-rich framework is build up from an extended π-conjugated carbon skeleton with porous structure and aromatic hydrogen (Ar-H) group, which endow it with high specific surface area, good electrochemical conductivity and a mass of active ion binding sites. Furthermore, the conducted 3D hierarchical porous structure provides a matrix for the hopping and transporting of electrons to ensure highly efficient charge collection and diffusion. Benefit from these, this free-standing flexible electrode can achieves a highly improved reversible and stable capacity of 1050 mAh g$^{-1}$ for LIBs which is higher than intrinsic carbon allotropes and organic molecules. The reversible capacity of SIBs is as high as 650 mAh g$^{-1}$ after 100 cycles which is the highest results among all the reported flexible electrodes as listed in our manuscript. The excellent cycle performance, high reversible capacity and superior rate capability make HsGDY as a promising candidate for LIBs and SIBs. This new anode material provides a new concept for the design of high performance flexible electrode.

## Methods

**HsGDY film preparation**. HsGDY was obtained following the synthetic route as shown in Fig. 1a. In a typical synthesis, 1.2592 g (4 mmol) of Tribromobenzene (Fig. 1a–1), 5.64 mL (40 mmol) of $(CH_3)_3SiC\equiv CH$, 0.2808 g (0.400 mmol) of $PdCl_2(PPh_3)_2$, 0.152 g (0.8 mmol) of CuI, and 0.2096 g (0.8 mmol) of $Ph_3P$ were added into 80 mL of TEA in this order. The mixture was stirred under a nitrogen atmosphere at 80 °C for 5 days in a tube sealing. After the solvent was evaporated, the residue was purified by column chromatography to yield Tris[(trimethylsilyl) ethynyl]benzene (Fig. 1a–2) as white powder.

To a solution of 48.7 mg (0.133 mmol) Tris[(trimethylsilyl)ethynyl]benzene in 15 mL THF was added 0.4 mL TBAF (1 M in THF, 0.4 mmol) and stirred at 6 °C for 30 min. The solution was then diluted with $CH_2Cl_2$ and washed with distilled water and dried with anhydrous $Na_2SO_4$. The solvent was removed in vacuum rotary evaporation to yield monomer (Fig. 1a–3).

This deprotected material was redituted with 25 mL pyridine and added slowly over 4 h to a solution of copper foils (The active surface area is 150 cm$^2$) in 50 mL pyridine at 60 °C. Then the mixture was stirred under a nitrogen atmosphere at 60 °C for 3 days. Upon completion copper foils were washed with acetone, N-Methyl pyrrolidone and calcined at 400 °C for 2 h in turn and a pale yellow film (Fig. 1a–4) was obtained on the copper foil.

**Characterization**. The samples were recorded on a Bruker D8 ADVANCE with Cu $K_\alpha$ radiation ($\lambda = 1.5406$ Å) at a scanning speed of 6° min$^{-1}$. Morphology details were examined using field emission scanning electron microscopy (FESEM, HITACHI S–4800) and transmission electron microscopy (TEM, HITACHI H–7650). The chemical structure of the products was characterized by Fourier transform infrared spectroscopy (FT-IR, Thermo-Fisher Nicolet iN10) and Raman spectroscopy (Thermo Scientific DXRxi, 532 nm). The X-Ray photoelectron spectrometer (XPS) was collected on VG Scientific ESCALab220i-XL X-Ray photoelectron spectrometer, using Al Kα radiation as the excitation sources. Nitrogen adsorption/desorption measurements were performed at 77 K using a Micromeritics ASAP2020 gas-sorption system. The UV–vis adsorption spectroscopy was recorded at HITACHI U–4100.

**Electrochemical analysis**. Electrochemical measurements were performed using CR2032 coin-type cells assembled in an argon-filled glovebox. The cells were assembled using the HsGDY as the cathode, a Li metal foil as the anode, a polypropylene separator (Celgard 2500), and a liquid electrolyte (ethylene carbonate, dimethyl carbonate, 1:1 by volume) with 1.0 M LiPF$_6$ for LIBs. For SIBs, Na metal, glass fiber, (ethylene carbonate, dimethyl carbonate, 1:1 by volume) with 1.0 M NaClO$_4$ and 5 wt% fluoroethylene carbonate (FEC) additive were used. The assembled half cells were cycled between 0.005 and 3 V using a LAND battery testing system. HsGDY electrode areas of 1 cm$^2$ with the thickness of (700 nm–2.9 μm) were used for electrochemical measurements in the form of free-standing. The active material is free-standing HsGDY film (0.1–0.25 mg) without any additions. The areal and volume loading density of the free-standing electrodes ranged from 0.1 to 0.25 mg cm$^{-2}$ and 1.43–0.8 g cm$^{-3}$. The high loading density (2.1 mg cm$^{-2}$) of HsGDY electrode is achieved by coating HsGDY powder on copper foil with 10% of Super P as conductive agent and 10% of polyvinylidene (PVDF) as binder. Electrochemical impedance spectroscopy (EIS) measurements were carried out using a ZHANER ZENNIUM electrochemical work station by applying an AC voltage of 5 mV amplitude at room temperature.

**DFT calculations**. All the calculations were performed under the framework of the dispersion-corrected density functional theory[56] (DFT–D3), as implemented in the VASP package with the projector-augmented-wave[57] (PAW) basis set and the PBE exchange-correlation functional within a generalized gradient approximation[58] (GGA). The kinetic energy cutoff was set to 600 eV. The monolayer model contains 2 monomers per unit cell and vacuum region of 20 Å in the z-direction. The geometry optimizations of the Li-adsorbed systems were carried out with the cell parameters remaining same to those of the pristine monolayer system. Uniform $5 \times 5 \times 1$ Gamma centered Monkhorst-Pack k-point meshes were employed for all systems. The convergence tolerance of the total energy was set to 0.0001 eV.

Any associated accession codes and references, are available in the Supplementary Information.

**Data availability**. The data that support the findings of this study are available from the corresponding author on request.

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

## Acknowledgements

This study was supported by the Hundred Talents Program and Frontier Science Research Project (QYZDB-SSW-JSC052) of the Chinese Academy of Sciences, the Natural Science Foundation of Shandong Province (China) for Distinguished Young Scholars (JQ201610) and the National Natural Science Foundation of China (21771187, 21790050 and 21790051). C.H. and J.H. acknowledge the discussion on language with X. Gao and W. B. Yifru.

## Author contributions

C.H. and Y.L. proposed the concepts and designed the experiments. C.H., N.W. and J.H. synthesized the samples. J.H., N.W., Z.C., H.D., L.F., Z.Y. and X.S. carried out the experiments. C.H., J.H., N.W., Z.C., H.D., L.F., Z.Y. and X.S. performed the analysis. Y.Y.,

Z.T., J.H. and N.W. conducted the DFT calculations. J.H. and C.H. wrote the manuscript with help from all co-authors and all authors contributed to interpretation of the data.

## Additional information

**Competing interests:** The authors declare no competing financial interests.

