## [Peer Review File · Nature Communications]

Reviewers' comments:

Reviewer #1 (Remarks to the Author):

This article reported a cross-coupling method to synthesize hydrogen substitutional graphdiyne (HsGDY) with highly porous structure. The physicochemical properties of materials were well characterized with several techniques such as NMR, XRD, FT-IR, XPS, Raman, TEM, SEM, XPS, and BET. Although the new electrode material shows good performance in both Li- and Na- ion batteries, this manuscript does not provide the important physical parameters of the electrodes, such as loading density and tap density. Without these parameters, it is not fair to compare the performance of the electrode with those of the other electrodes. The reviewer also cannot evaluate true potential of the electrode for LIB or NIB applications. With these reasons, I do not recommend publication of the current form of the manuscript in Nature Communications. The authors may resubmit the revised manuscript to Nature Communications after addressing the below comments.

1) First of all, the authors should provide the density of the new electrode materials and the loading density (mg/cm²) for the electrochemical measurements. The high rate performance of the electrode is mainly attributed to the ultrathin feature of the free-standing films (~960 nm). As both specific capacity and rate performance significantly depend on the loading density of the electrode materials, the authors need to show loading dependent capacity/rate-performance for this new material. It is recommended to show the superior performances at high loading density. In addition, the comparison table 1 and 2 also must include the loading density information for fair comparisons.

2) Based on the mechanism in Figure 1, no oxygen should be introduced in HsGDY. However, the XPS results showed a high oxygen ratio in HsGDY. Please give a discussion about the origin of the oxygen. The XPS shows the existence of C-O C=O bonds through the peak fitting of C1s (Figure 1F), but no corresponding peaks could be observed in FTIR measurement. What will be the role of oxygen functional groups? More defects for Li or Na ion storage?

3) The authors proposed several possible mechanisms for the Li- or Na- ion storage. The reviewer cannot clearly see the differences of CV shapes with those of the reduced GO or other high surface carbon materials.

4) The figures numbers should be consistent. Please check the Figure 2D-G, which should be Figure 1D-G

Reviewer #2 (Remarks to the Author):

This manuscript shows the electrochemical performance of the HsGDY for LIBs and SIBs. They also fabricated the 3D HsGDY. Searching for new LIB and SIB anode is important for improving performance of LIBs by replacing graphite. Recently, many studies have been done to search for new LIBs or SIBs. So far it is not yet successful. However, in this manuscript, the gravimetric capacity of HsGDY is pretty high, ~1000 and 600 mAh/g compared with that in different materials for LIBs and SIBs, which is greater than the value (372 mAh/g) of graphite. Other performance such as the Coulombic efficiency is also pretty high. These results show a possibility of the HsGDY for Li and Na ions batteries. Some comments arise below.

(1) What is the volumetric capacity? Greater than the capacity (~800 mAh/cm³) of graphite?

(2) There are a lot of phases in GDY. What is the phase in your study? Alpha-GDY? or Gamma-GDY?

The schematic in your manuscript shows gamma-GDY. It is well known that gamma-GDY is most stable. The phase information is necessary because the capacity depends on what phases are.

(3) Please discuss the capacity and performance compared with multilayer GDY ([Appl. Phys. Lett. 103, 263904 (2013)]) and bulk GDY [Ref. 16]. What is the difference? Why the capacity is lower or higher than those? Why hydrogen substitution makes a big difference?

(4) Why Li and Na give pretty different capacity?

Reviewer #3 (Remarks to the Author):

This paper proposes a new material, hydrogen substituted graphdiyne, as an electrode materials for Li and Na-ion batteries. The material can be synthesized on a copper substrate and it has been characterized and tested, showing much promise. Key properties of the material are that it is relatively easy to fabricate, it is transparent, it has good performance characteristics as a material for alkali metal ion batteries and it is flexible. Due to the interest in energy storage materials, I believe that this work will be of significant interest to the general field and the community.

Some specific comments:

(i) Due to the somewhat heterogeneous nature of the material at the mesoscale, it would be of interest to know how reproducible the morphology, and particularly the performance results are. In all cases, it seems that a single sample was considered and results given. Are these typical results, the only results or the best results?

(ii) The way in which capacities are calculated should be clearly defined (does the total mass include the metal ions or not?) Both methods are used in the literature and it is important to clarify so materials can be properly compared.

(iii) The authors have cited their work on graphdiyne for lithium and sodium batteries (14-19) - sodium should be mentioned as well as lithium on line 57.

(iv) In their discussion on LIBs, the authors discuss the preferred binding sites of Li in graphdiyne. These are slightly different from those in for Na in graphdiyne so it could be interesting to discuss this point and its relevance to the new material in the section on SIBs (eg. Xu et al. RSC Adv, 6 (2016) 25594; Farokh Niaei et al. J Power Sources, 343, (2017) 354; Zhang et al. J Mat Chem A, 5, (2017) 2045).

(v) A significant concern about the paper as it currently appears is the English expression. Even the title describing the material as "Hydrogen substitutional graphdiyne" is not a clear choice. In some places, this causes difficulty in understanding the content of the manuscript. Similarly, the paper and the Figures should be carefully proof-read (eg Figure 1A, 'coupling' is spelled incorrectly, and in Figure 2g, 'relative' is incorrectly spelled on the axis label.)

Overall, the work is interesting, timely and seems to be solid however the presentation needs to be improved before it is publishable in Nature Communications.

Reviewers' comments:

Reviewer #1 (Remarks to the Author):

The authors fully addressed my concerns and made significant improvements.

Reviewer #2

This reviewer recommended the manuscript for publication in Nature Communications in his/her confidential comments to Editor.

Reviewer #3 (Remarks to the Author):

The authors have carried out substantial additional work to address the referee comments and concerns. In most cases this has improved the manuscript however, there are some additional concerns raised.

1. On page 5, the discussion of the phase of HsGDY is confusing. A comment pointing out that various phases of GDY exist, and that the HsGDY can be considered as a H-substituted γ -GDY would be sufficient. I don't think it makes sense to say that it should be classed as γ -GDY rather than α -GDY or β -GDY.
2. The authors now say that the NMR and FTIR spectra show no evidence of C-O and C=O bonds. This does not seem to be correct. In FTIR of graphene oxides, peaks at about 3400 cm^{-1} are typically assigned as OH stretch; at about 1700 as C=O stretch; at about 1100 and 1400 as C-O stretch and all these are present and unassigned in the FTIR in Fig 1h. Similarly, the peak marked defect in Fig 1g would be where presence of O would be observed, and the NMR is not inconsistent with typical spectra of graphene oxides. Therefore, it seems likely that there is a significant amount of O present in the material and the implication of this on the results and characterization of the material needs to be considered and discussed.
3. Stating a loading density for MCOF to be 1 mg does not make sense and it must be given per volume or area.
4. Equation (1) does not show the mechanism of the lithium storage and is not useful. This should be removed. The discussion below that is more useful although it should be justified more thoroughly.
5. On page 13 it is stated that the definition of binding energy is given in the Supplementary information but this not the case. More details of the theoretical calculations and results are needed. The units should be given in 15(b). Is the result in 15(d) the total binding energy of all Li or per atom? How are the configurations in 15(b) and 15(d) selected and how is the storage capacity determined? Similar comments apply to 16(b).
6. The paper still has problems associated with the English expression which make it hard to understand in places. A few examples are: on page 6, the sentence "Compared with that of triethynylbenzene monomer (Supplementary Fig. 5), Raman spectrum of HsGDY shows increasing intensity of graphitic C=C stretching vibration in G-band." and "the high degree of π -conjugated system which"; on page 8 "The loading density of the film is weighed about 0.10-0.11 mg cm^{-2} when the thickness is about 700 nm" These are other issues need to be fixed.

REVIEWERS' COMMENTS:

Reviewer #3 (Remarks to the Author):

The authors have modified their manuscript in response to the comments of the referees.

Point by Point Responses:

Reviewer #1: This article reported a cross-coupling method to synthesize hydrogen substitutional graphdiyne (HsGDY) with highly porous structure. The physicochemical properties of materials were well characterized with several techniques such as NMR, XRD, FT-IR, XPS, Raman, TEM, SEM, XPS, and BET. Although the new electrode material shows good performance in both Li- and Na- ion batteries, this manuscript does not provide the important physical parameters of the electrodes, such as loading density and tap density. Without these parameters, it is not fair to compare the performance of the electrode with those of the other electrodes. The reviewer also cannot evaluate true potential of the electrode for LIB or NIB applications. With these reasons, I do not recommend publication of the current form of the manuscript in Nature Communications. The authors may resubmit the revised manuscript to Nature Communications after addressing the below comments.

Q1: First of all, the authors should provide the density of the new electrode materials and the loading density (mg/cm²) for the electrochemical measurements. The high rate performance of the electrode is mainly attributed to the ultrathin feature of the free-standing films (~960 nm). As both specific capacity and rate performance significantly depend on the loading density of the electrode materials, the authors need to show loading dependent capacity/rate-performance for this new material. It is recommended to show the superior performances at high loading density. In addition, the comparison table 1 and 2 also must include the loading density information for fair comparisons.

A: Thanks for your advice. The loading density of the HsGDY electrodes are provided in table 1 and 2 of our revision. The loading density of our sample is 1.43 g cm⁻³ (volume density) and 0.1 mg cm⁻² (areal density). The loading density information of the references listed in table 1 and 2 are also carefully investigated.

However, not all of the information can be found by checking the related publication of the references. The details are listed as below:

In table 1, Ref. 33, ref. 35, ref. 37, ref. 39 and ref. 40 didn't mention the loading density information of the electrodes. Ref. 34 reported that the loading density and thickness of the LBL-MWNT electrodes are 0.83 g cm^{-3} and $0.3 \text{ }\mu\text{m}$, respectively. Ref. 36 reported that the loading density of pristine graphene is $20.8\text{-}24.2 \text{ mg cm}^{-3}$ while its thickness is not mentioned. Ref. 38 reported that the loading density of carbon nanotube films is 0.796 mg cm^{-2} . Ref. 41 reported that the loading mass of the mesoporous covalent organic frameworks is approximately 1 mg.

In table 2, Ref. 49, ref. 48, ref. 30, ref. 51 and ref. 53 didn't mention the quality or thickness of the electrode. The loading areal density (or volume density) of the electrodes in ref. 47, ref. 50, ref. 52 and ref. 29 are 4.3 mg cm^{-2} (or 0.96 mg cm^{-3}), 1 mg cm^{-2} (or 0.22 mg cm^{-3}), 0.64 mg cm^{-2} (or 0.14 mg cm^{-3}) and 1 mg cm^{-2} , respectively.

Compared with those non-flexible electrodes which are coated on copper foil, free-standing HsGDY film exhibits a lower areal density because of the thin thickness. However, the volume density of HsGDY is higher than all the listed materials. In most cases, the thickness of free-standing electrodes is always thin in order to enhance the flexibility such as layer-by-layer MWNT electrodes (ref. 34).

Meanwhile, in the revision, we also gave the electrochemical performance results of HsGDY with different thickness and loading density which were studied the same time as our paper reported results. We have tried to control the thickness of the film by varying the initial amount of the monomer. All of the experimental data is repeated for at least five times. We could obtain higher areal loading density (0.2 mg cm^{-2} and 0.25 mg cm^{-2}) than that in our paper by preparing thick films, but those as-prepared HsGDY films are not chosen because of poor quality and low tap density (0.8 g cm^{-3} and 0.86 g cm^{-3}). However, this drawback may be overcome through a different catalytic process which we are ongoing to study. We actually had haven plan to report

the systematic study of electrochemical performance of HsGDY with different thickness after this paper.

On the other hand, high loading density (2.1 mg cm^{-2}) of HsGDY electrode is achieved by coating HsGDY powder on copper foil with 10% of Super P as conductive agent and 10% of PVDF as binder. The powder was obtained by grinding the HsGDY film. The electrodes exhibit a comparative capacity with that of free-standing HsGDY film (thickness of 700 nm) as shown below (Supplementary Fig. 14).

The morphology and rate-performance with different loading density for HsGDY material are discussed in our revised manuscript as below:

Page 8 line 196-212:

Moreover, the thickness and areal density of the HsGDY films can be tuned by controlling the initial amount of monomer (the preparation details were described in Supplementary Information). As shown in Supplementary Fig. 9f, the thickness of the film is $2.5 \text{ }\mu\text{m}$ when the initial amount of monomer is 0.266 mmol (twice the weight of the original HsGDY film with 700 nm thickness). The loading density of electrode based on this HsGDY film increases to $0.18\text{-}0.20 \text{ mg cm}^{-2}$. In the meantime, the pore size become larger and the film become more porous (Supplementary Fig. 9d and e). This phenomenon can be ascribed to the catalytic reaction which is discussed earlier in the article. However, the thickness of the film as grown is changed slightly (only $2.9 \text{ }\mu\text{m}$) when the amount of monomer is threefold the weight of the monomer preparing HsGDY film with 700 nm thickness (Supplementary Fig. 9i). The loading density of electrode based on this HsGDY film is about $0.25\text{-}0.26 \text{ mg cm}^{-2}$. Since the copper works as both the support and catalyst, the thicker the film grows the more difficult the catalytic reaction occurs. It also can be seen from Supplementary Fig. 9g and h, the porous film collapse and the pore become smaller than the other two films. All the volume densities (tap density) of the HsGDY film with thickness of 2.5 and $2.9 \text{ }\mu\text{m}$ are smaller than that of HsGDY film with thickness of 700 nm .

Supplementary Figure 9. The morphology of the HsGDY film synthesized by different amount of monomer. (a-c) 0.133 mmol, (d-f) 0.266 mmol, (g-i) 0.399 mmol.

Page 12 line 278-288:

Meanwhile, the electrochemical performance of HsGDY films with different density mentioned above was also measured. As shown in Supplementary Fig. 12, the reversible capacity and rate performance of HsGDY film in LIBs are slightly reduced with the increasing of areal density. Those can be ascribed to the low tap density and poor quality of the thick films. On the other hand, high loading density (2.1 mg cm^{-2}) of HsGDY electrode is achieved by coating HsGDY powder on copper foil with 10% of Super P as conductive agent and 10% of PVDF as binder (the preparation detail can be seen in Supplementary Information). The powder was obtained by grinding the HsGDY film. The electrodes exhibit a comparative capacity with that of free-standing HsGDY film (Supplementary Fig. 14a). But this kind of HsGDY based electrode was not free-standing, bendable and transparent, comparing with HsGDY film based electrode.

Similar results as LIBs have been obtained while applied the HsGDY films with other different density for SIBs (Supplementary Fig. 13 and Supplementary Fig. 14b).

Supplementary Figure 12. Rate performance of the HsGDY electrodes synthesized by different amount of monomer in LIBs, (a) 0.133 mmol, (b) 0.266 mmol, (c) 0.399 mmol.

Supplementary Figure 13. Rate performance of the HsGDY electrodes synthesized by different amount of monomer in SIBs, (a) 0.133 mmol, (b) 0.266 mmol, (c) 0.399 mmol.

Supplementary Figure 14. Rate performance of the HsGDY electrodes coated on copper foil, (a) in LIBs, (b) in SIBs.

Q2: Based on the mechanism in Figure 1, no oxygen should be introduced in HsGDY. However, the XPS results showed a high oxygen ratio in HsGDY. Please give a discussion about the origin of the oxygen. The XPS shows the existence of C-O C=O

bonds through the peak fitting of C1s (Figure 1F), but no corresponding peaks could be observed in FTIR measurement. What will be the role of oxygen functional groups? More defects for Li or Na ion storage?

A: Thanks for your comment. As showed in our paper, the existence of oxygen is clearly observed in the XPS spectrum of HsGDY (Fig. 1f and Supplementary Fig. 3). However, no distinguishable carbon-oxygen bond can be found in the solid-state NMR and other characterization methods (such as Raman and FTIR measurement). As a surface analysis tool, XPS spectrum can only detect the components on the surface with depth in no more than 10 nm. In consideration of the high specific surface area and the existed defects of HsGDY samples, the origin of the C-O and C=O bonds might be ascribed to the chemical adsorption of oxygen on the surface of HsGDY or the reaction between oxygen and some terminated acetylenic bond exposed on the surface of the HsGDY films, which is also reported by the references (ref. 15 and ref. 25). The solid-state NMR, which got signal from the bulk phase of HsGDY, clearly evidence the lack or little amount of C-O and C=O bonds from the very weak peak marked as below. All those might be the reason that the XPS shows the existence of C-O C=O bonds, but no corresponding peaks could be observed in FTIR measurement. Although, as the reviewer said, the oxygen functional groups are reported to store Li or Na ion in many references (ref. 6, ref. 28, and ref. 50), the role of them in electrochemical process of our HsGDY electrode is not the main part owing to their little amount.

We added some discussion in our revision as below in Page 5 line 119-126:

The XPS shows the existence of C-O C=O bonds through the peak fitting of C1s (Fig. 1f), but no corresponding peaks could be clearly observed in Fourier transform infrared spectroscopy (FT-IR) measurement and solid-state NMR (Fig. 1d), indicating the little amount of C-O and C=O bonds exist. The origin of the C-O and C=O bonds might be ascribed to the chemical adsorption of oxygen on the surface of HsGDY or

the reaction between oxygen and some terminal acetylenic bond exposed on the surface of the HsGDY films, which is also reported by the references^{15,25}.

Figure R1. The solid-state NMR of HsGDY sample.

Q3: The authors proposed several possible mechanisms for the Li- or Na- ion storage. The reviewer cannot clearly see the differences of CV shapes with those of the reduced GO or other high surface carbon materials.

A: Thanks for your comment. We have modified the discussion of the electrochemical process with more detailed description to explain the differences of CV shapes. We added those discussion in our revision (page 12 line 289-324) as below:

Li storage mechanism in HsGDY. According to the obtained reversible capacity, the lithium storage mechanism can be described as the following equation:

Three main storage modes in the equation are proposed based on previous reported works. As marked in Fig. 3e, (i) Li atoms can be bound near by the H atoms in this hydrocarbon. This always causes changes in the defined atom positions of the C and

H atoms; this bonding distortion would be an activated process, which might result in hysteresis²¹. The substantial work that has been done on ternary graphite-alkali-hydrogen materials showed that charge transfer from alkalis to hydrogen in carbons is expected. Although the Li storage mechanism for various hydrogen-terminated carbon materials is still unclear, many references have reported that lithium atom could bind to H atom at about 0.7-1.5 V which is in accordance with the inconspicuous plateau in cyclic voltammogram (CV) and charge-discharge curves (Fig. 3a and Supplementary Fig. 10a)⁴². (ii) Li atoms can be adsorbed above the center of 6-C hexagon^{21,43}. This Li storage mode exists in most graphite electrode. (iii) Li atoms can be located at the vicinity of butadiyne in the hexagonal pore^{43,44}. It is also reported that high specific capacity is obtained by storage of lithium in micropores at a very low potential⁴⁵. In this work, HsGDY delivers considerable capacity at the plateau of near 0 V observed in CV and charge-discharge curves (Fig. 3a and Supplementary Fig. 10a). This can be attributed to the insertion of Li above the benzene ring (mode ii) and the storage of Li in the micropores with main size distribution of 0.7 nm (mode iii). The inconspicuous peaks in the CV of the sample imply that the capacity is mainly dominated by faradaic pseudocapacitance rather than redox reaction because of the high specific surface area.

In the meantime, the theoretical calculations of Li storage were performed using a first-principles method based on density functional theory. The definition of binding energy and the binding energy of single Li atom at different adsorption sites on selected repeating unit can be found in Supplementary Fig. 15a-b. As shown in Supplementary Fig. 15c-d, the stabilities of Li₂₈-C₂₄H₆ complex was examined by their binding energies on single layer HsGDY. It can be observed that the Li atoms are stored at the nearby of H atoms, benzene ring and butadiyne in calculation, which are in accordance with that in reference^{21,42-45}. Moreover, the Li storage capacity is calculated to be 2553 mAh g⁻¹ in which the adsorption in both sides of HsGDY layer is required. That the calculated capacity is larger than experimental result can be

ascribed to the omission of steric hindrance between the layers in AB-stacking multilayer HsGDY.

Supplementary Figure 15. The calculated binding energy of Li atoms on selected repeating unit, (a-b) single Li atom at different sites, (c) top view and (d) cross-section view of rational Li₂₈-C₂₄H₆ complex and its binding energy.

Q4: The figures numbers should be consistent. Please check the Figure 2D-G, which should be Figure 1D-G.

A: Thanks for your kindly comment. We have corrected the figure number as you point out, and checked the other figures numbers of the whole manuscript carefully.

Reviewer #2: This manuscript shows the electrochemical performance of the HsGDY for LIBs and SIBs. They also fabricated the 3D HsGDY. Searching for new LIB and SIB anode is important for improving performance of LIBs by replacing graphite. Recently, many studies have been done to search for new LIBs or SIBs. So far it is not yet successful. However, in this manuscript, the gravimetric capacity of HsGDY is pretty high, ~1000 and 600 mAh/g compared with that in different materials for

LIBs and SIBs, which is greater than the value (372 mAh/g) of graphite. Other performance such as the Coulombic efficiency is also pretty high. These results show a possibility of the HsGDY for Li and Na ions batteries. Some comments arise below.

Q1: What is the volumetric capacity? Greater than the capacity (~800 mAh/cm³) of graphite?

A: We have added the volumetric capacity in rate performance of HsGDY in Fig. 3a. The volumetric capacity of HsGDY is 1447 mAh cm⁻³ at the current density of 0.1 A g⁻¹, and can even achieve 815 mAh cm⁻³ while the current density increased to 5 A g⁻¹, which are obviously greater than the capacity (~800 mAh cm⁻³) of graphite. Meanwhile, the volumetric capacity of HsGDY in comparison of graphite is also discussed in the revised text (page 11 line 249-255) as below:

The reversible gravimetric capacity and volumetric capacity of the HsGDY are 1012 mAh g⁻¹ and 1447 mAh cm⁻³ at the current density of 0.1 A g⁻¹, and can even achieve 570 mAh g⁻¹ and 815 mAh cm⁻³ while the current density increased to 5 A g⁻¹. These experimental capacities is much greater than the theoretical gravimetric and volumetric capacities of 372 mAh g⁻¹/818 mAh cm⁻³ and 744 mAh g⁻¹ for graphite and γ -graphdiyne suggesting that HsGDY can serve as a promising high-capacity lithium ion battery anode.

Supplementary Figure 12a. Rate performance of the HsGDY in LIBs.

Supplementary Figure 13a. Rate performance of the HsGDY electrodes in SIBs.

Q2: There are a lot of phases in GDY. What is the phase in your study? Alpha-GDY? or Gamma-GDY? The schematic in your manuscript shows gamma-GDY. It is well

know that gamma-GDY is most stable. The phase information is necessary because the capacity depends on what phases are.

A: A lot of phases in GDY have been reported by the references including Alpha-GDY and Gamma-GDY. In this work, HsGDY was synthesized based on the structure of gamma-GDY, but with different initial monomer. The monomer of gamma-GDY is hexaethynylbenzene, while that of HsGDY is triethynylbenzene. Actually, we concentrate on the molecule design for improving the electrochemical performance of carbon based materials. Hence, the synthesized material HsGDY would like a new carbon-rich material rather than a new phase of GDY. If it is necessary for us to provide the phase information, the HsGDY may be attributed to gamma-GDY.

We added those discussions in our revision (page 5 line 100-104) as below:

Since HsGDY was synthesized based on the structure of γ -GDY, but with different initial monomer, the synthesized material HsGDY would like a new carbon-rich material rather than a new phase of GDY. In consideration of the important role of phase information in Li storage capacity, the HsGDY may be classed as γ -GDY rather than α -GDY or β -GDY^{15,23,24}.

Q3: Please discuss the capacity and performance compared with multilayer GDY ([Appl. Phys. Lett. 103, 263904 (2013)]) and bulk GDY [Ref. 16]. What is the difference? Why the capacity is lower or higher than those? Why hydrogen substitution makes a big difference?

A: Thanks for your suggestion. Recently, many efforts have been made to improve the electrical performance of GDY. [Appl. Phys. Lett. 103, 263904 (2013)] (as **Ref. 29** in our manuscript) have reported that multilayer α -graphdiyne can serve as a promising high-capacity lithium ion battery anode in which the theoretical specific/volumetric capacities can reach up to 2719 mAh g⁻¹/2032 mAh cm⁻³ using the

first-principles calculations. On the other hand, bulk γ -graphdiyne [Ref. 16] with theoretical capacity of 744 mAh/g is reported. Here, a new carbon-rich material is synthesized. The theoretical capacity and voltage of Li storage is calculated through DFT. The difference between HsGDY and the others is discussed. We added those discussions in our revision (page 13 line 313-332) as below:

In the meantime, the theoretical calculations of Li storage were performed using a first-principles method based on density functional theory. The definition of binding energy and the binding energy of single Li atom at different adsorption sites on selected repeating unit can be found in Supplementary Fig. 15a-b. As shown in Supplementary Fig. 15c-d, the stabilities of $\text{Li}_{28}\text{-C}_{24}\text{H}_6$ complex was examined by their binding energies on single layer HsGDY. It can be observed that the Li atoms are stored at the nearby of H atoms, benzene ring and butadiyne in calculation, which are in accordance with that in reference^{21,42-45}. Moreover, the Li storage capacity is calculated to be 2553 mAh g⁻¹ in which the adsorption in both sides of HsGDY layer is required. That the calculated capacity is larger than experimental result can be ascribed to the omission of steric hindrance between the layers in AB-stacking multilayer HsGDY.

The calculated Li storage capacity of HsGDY is lower than that of α -graphdiyne and higher than that of γ -graphdiyne. This is because α -graphdiyne is mainly comprised by carbyne which always shows much higher capacity than benzene ring. Hence, the theoretical capacity of α -graphdiyne may be highest among 2D layers of sp-sp² hybrid carbon networks. On the other hand, hydrogen is introduced into γ -graphdiyne to stabilize the structure, enlarge the pore size and provide more active binding sites. Therefore, as far as we know, HsGDY delivers the highest experimental capacity among the synthesized sp-sp² hybrid carbon networks.

Supplementary Figure 15. The calculated binding energy of Li atoms on selected repeating unit, (a-b) single Li atom at different sites, (c) top view and (d) cross-section view of rational $\text{Li}_{28}\text{-C}_{24}\text{H}_6$ complex and its binding energy.

Q4: Why Li and Na give pretty different capacity?

A: In most references, the Na storage capacity of the electrodes is smaller than the Li storage capacity of them due to the larger ion radius of Na^+ (1.02 \AA) than that of Li^+ (0.76 \AA) and sluggish kinetic of Na diffusion.

We have proposed the Na storage mechanism in the revised manuscript (page 17 line 378-385) as below:

To further understand the Na storage in HsGDY electrodes, rational configuration of HsGDY as shown in Supplementary Fig. 16 was selected for the theoretical calculation of Na storage. It can be found that a lower binding energy was got in the

optimized $\text{Na}_{22}\text{-C}_{24}\text{H}_6$ configuration in comparison with $\text{Li}_{28}\text{-C}_{24}\text{H}_6$ configuration. It can be attributed to that strong repulsion among Na atoms and the large diameter make the substantial storage of Na in hexagonal pore difficult. This phenomenon is also observed in other carbon materials with many micropores²².

Supplementary Figure 16. The calculated binding energy of Na atoms on selected repeating unit, (a) top view and (b) cross-section view of rational $\text{Na}_{22}\text{-C}_{24}\text{H}_6$ complex and its binding energy.

Reviewer #3: This paper proposes a new material, hydrogen substituted graphdiyne, as an electrode materials for Li and Na-ion batteries. The material can be synthesized on a copper substrate and it has been characterized and tested, showing much promise. Key properties of the material are that it is relatively easy to fabricate, it is transparent, it has good performance characteristics as a material for alkali metal ion batteries and it is flexible. Due to the interest in energy storage materials, I believe that this work will be of significant interest to the general field and the community.

Q1: Due to the somewhat heterogeneous nature of the material at the mesoscale, it would be of interest to know how reproducible the morphology, and particularly the

performance results are. In all cases, it seems that a single sample was considered and results given. Are these typical results, the only results or the best results?

A: We agree with this comment of reviewer. We are sure about that all of the morphology and electrochemical performance are reproducible in our manuscript. All of the experimental data is repeated again and again in our experiments (at least three times). Herein, the repeated experiment data of three times are provided in Supplementary Fig. 8 and Supplementary Fig. 11. When the loading density of the HsGDYs is almost the same, the morphology and electrochemical performance are similar with each other. Meanwhile, we also give the electrochemical performance for different samples of HsGDY. All of the experimental data demonstrate the good reproducibility of HsGDY samples.

We added those discussion in our revision (page 8 line 192-196) as below:

The loading density of the film is weighed about $0.1\text{-}0.11\text{ mg cm}^{-2}$ when the thickness is about 700 nm. The morphology of HsGDY film is similar with each other for different synthetic batches which demonstrate the good reproducibility of carbon-rich framework (Supplementary Fig. 8).

Supplementary Figure 8. The reproducibility of the HsGDY film. (a-c) sample 1, (d-f) sample 2, (g-i) sample 3.

The reproducibility of the electrochemical performance for HsGDY electrodes in LIBs is investigated in Supplementary Fig. 11a-c. It is observed that the rate performance and cycle performance of the different HsGDY film is almost same. The similar phenomenon is found in SIBs as shown in Supplementary Fig. 11d-f. (Page 12 line 274-278)

Supplementary Figure 11. The reproducibility of the electrochemical performance for HsGDY electrodes, (a-c) in LIBs, (d-f) in SIBs.

Q2: The way in which capacities are calculated should be clearly defined (does the total mass include the metal ions or not?) Both methods are used in the literature and it is important to clarify so materials can be properly compared.

A: Thanks for your comment. The electrode is described more detailed. The active material is free-standing HsGDY film (0.1-0.25 mg) without any additions. At the same time, the capacity of the electrode is analyzed by both experimental and theoretical method. The metal ions are not observed in the characterizations which imply the high purity of the synthesized material. Hence, the role of the metal ions is

not discussed in the manuscript. The comparison with other carbon materials is listed in table 1 and 2.

We added the detailed information about our electrode in our supporting information as below:

HsGDY electrode areas of 1 cm^2 with the thickness of (700 nm-2.9 μm) were used for electrochemical measurements in the form of. The active material is free-standing HsGDY film (0.1-0.25 mg) without any additions. The areal and volume loading density of the free-standing electrodes ranged from 0.1 to 0.25 mg cm^{-2} and 1.43-0.8 g cm^{-3} .

Q3: The authors have cited their work on graphdiyne for lithium and sodium batteries (14-19) - sodium should be mentioned as well as lithium on line 57.

A: Thanks for your suggestion. The proper revision is added as below (Page 2 line 54-57).

While recently, our group is pioneer in developing new carbon allotropes graphdiyne as a high capacity electrode for LIBs and sodium ion batteries (SIBs), which gives new insight into the layered material electrodes¹⁴⁻¹⁹.

Q4: In their discussion on LIBs, the authors discuss the preferred binding sites of Li in graphdiyne. These are slightly different from those in for Na in graphdiyne so it could be interesting to discuss this point and its relevance to the new material in the section on SIBs (eg. Xu et al. RSC Adv, 6 (2016) 25594; Farokh Niaei et al. J Power Sources, 343, (2017) 354; Zhang et al. J Mat Chem A, 5, (2017) 2045).

A: Thanks for your advice. The Li and Na storage in HsGDY is discussed based on the references and calculations.

We added those discussions in our revision as below:

(page 13 line 313-324)

In the meantime, the theoretical calculations of Li storage were performed using a first-principles method based on density functional theory. The definition of binding energy and the binding energy of single Li atom at different adsorption sites on selected repeating unit can be found in Supplementary Fig. 15a-b. As shown in Supplementary Fig. 15c-d, the stabilities of $\text{Li}_{28}\text{-C}_{24}\text{H}_6$ complex was examined by their binding energies on single layer HsGDY. It can be observed that the Li atoms are stored at the nearby of H atoms, benzene ring and butadiyne in calculation, which are in accordance with that in reference^{21,42-45}. Moreover, the Li storage capacity is calculated to be 2553 mAh g^{-1} in which the adsorption in both sides of HsGDY layer is required. That the calculated capacity is larger than experimental result can be ascribed to the omission of steric hindrance between the layers in AB-stacking multilayer HsGDY.

Supplementary Figure 15. The calculated binding energy of Li atoms on selected repeating unit, (a-b) single Li atom at different sites, (c) top view and (d) cross-section view of rational $\text{Li}_{28}\text{-C}_{24}\text{H}_6$ complex and its binding energy.

(page 17 line 378-385):

To further understand the Na storage in HsGDY electrodes, rational configuration of HsGDY as shown in Supplementary Fig. 16 was selected for the theoretical calculation of Na storage^{18,54,55}. It can be found that a lower binding energy was got in the optimized Na₂₂-C₂₄H₆ configuration in comparison with Li₂₈-C₂₄H₆ configuration. It can be attributed to that strong repulsion among Na atoms and the large diameter make the substantial storage of Na in hexagonal pore difficult. This phenomenon is also observed in other carbon materials with many micropores²².

Supplementary Figure 16. The calculated binding energy of Na atoms on selected repeating unit, (a) top view and (b) cross-section view of rational Na₂₂-C₂₄H₆ complex and its binding energy.

Q5: A significant concern about the paper as it currently appears is the English expression. Even the title describing the material as "Hydrogen substitutional graphdiyne" is not a clear choice. In some places, this causes difficult in understanding the content of the manuscript. Similarly, the paper and the Figures should be carefully proof-read (eg Figure 1A, 'coupling' is spelled incorrectly, and in Figure 2g, 'relative' is incorrectly spelled on the axis label.)

Overall, the work is interesting, timely and seems to be solid however the presentation needs to be improved before it is publishable in Nature Communications.

A: We appreciate your positive comment. The whole manuscript has been revised carefully. In this work, HsGDY was synthesized based on the structure of GDY, but with different initial monomer. The monomer of GDY is hexaethynylbenzene, while that of HsGDY is triethynylbenzene in which three ethynyl was replaced by hydrogen. Hence, we describe the new material as "Hydrogen substituted graphdiyne". We changed our paper title from “**Hydrogen substitutional graphdiyne as carbon-rich flexible electrode for lithium and sodium ion batteries**” to “**Hydrogen substituted graphdiyne as carbon-rich flexible electrode for lithium and sodium ion batteries**”. We also carefully improved the language of our presentation.

Point by Point Responses:

Reviewer #1: The authors fully addressed my concerns and made significant improvements.

A: We thank the reviewer for constructive suggestions and taking the time to read our revision carefully.

Reviewer #2: This reviewer recommended the manuscript for publication in Nature Communications in his/her confidential comments to Editor.

A: We are very grateful for all the comments and suggestion from the reviewer.

Reviewer #3: The authors have carried out substantial additional work to address the referee comments and concerns. In most cases this has improved the manuscript however, there are some additional concerns raised.

1. On page 5, the discussion of the phase of HsGDY is confusing. A comment pointing out that various phases of GDY exist, and that the HsGDY can be considered as a H-substituted γ -GDY would be sufficient. I don't think it makes sense to say that it should be classed as γ -GDY rather than α -GDY or β -GDY.

A: Thanks for your suggestion. We have revised γ -GDY to hydrogen substituted γ -GDY in our revision.

2. The authors now say that the NMR and FTIR spectra show no evidence of C-O and C=O bonds. This does not seem to be correct. In FTIR of graphene oxides, peaks at about 3400 cm^{-1} are typically assigned as OH stretch; at about 1700 as C=O stretch; at about 1100 and 1400 as C-O stretch and all these are present and unassigned in the FTIR in Fig 1h. Similarly, the peak marked defect in Fig 1g would be where presence of O would be observed, and the NMR is not inconsistent with typical spectra of graphene oxides. Therefore, it seems likely that there is a significant amount of O present in the material and the implication of this on the results and characterization

of the material needs to be considered and discussed.

A: Thanks for your comment. We agree that we should describe the existence of C-O and C=O bonds with a strict expression. Actually, in our last revision we also admit the small amount of C-O and C=O bonds exist (The discussion in our last revision was “The XPS shows the existence of C-O C=O bonds through the peak fitting of C1s (Fig. 1f), but no corresponding peaks could be clearly observed in Fourier transform infrared spectroscopy (FT-IR) measurement and solid-state NMR (Fig. 1d), indicating the little amount of C-O and C=O bonds exist”). But as the reviewer said, Fig. 1g shows the peak around 1100 cm^{-1} for C-O and the peak at 1700 cm^{-1} for C=O. We modified the related discussion about IR part. While as a qualitative analysis tool, the signal of FTIR spectrum depends sensitively on the dipole moment variety of chemical bond stretch. The dipole moment variety of C-O or C=O bond is stronger than that of C=C or $\text{-C}\equiv\text{C-C}\equiv\text{C-}$ bond. Hence, FTIR spectrum can't exactly describe the ratio of oxygen in the HsGDY bulk. NMR spectrum as a bulk analysis tool revealed a real content of carbon in C=C, $\text{-C}\equiv\text{C-C}\equiv\text{C-}$, C-O and C=O bonds, in which the signal strength of the peak stands for the relative amount of each carbon species (Figure R1a). If there are significant amounts of O present in the material, the featured C atoms of C-O and C=O should be clearly observed in NMR spectra. In Fig. 1d, the peaks at around 67.5 and 166.0 ppm correspond to the C atoms of C-O and C=O bonds, respectively (Science 2008, 321, 1815-1817; Eur. Polym. J. 2012, 48, 705-711). Those peaks are not strong and clear compared with those of C=C and $\text{-C}\equiv\text{C-C}\equiv\text{C-}$ bonds. These results suggested that the amount of O in HsGDY is very small. The origin of O might be ascribed to the chemical adsorption of oxygen on the surface of HsGDY, as well as the reaction between oxygen and some exposed terminal or side acetylenic bonds on the surface of HsGDY. The chemical adsorption of oxygen on the surface and reaction of O with side vinyl have been reported to be existed unavoidably in many carbon materials including graphene and carbon nanofiber (Nano Lett. 2009, 9, 1752-1758); Angew. Chem. Int. Ed. 2014, 53, 6905-6909); Adv. Mater. 2013, 25, 250-255). Although the oxygen functional groups

are reported to store Li or Na ion in many references (ref. 6, ref. 28, and ref. 50), the role of them in electrochemical process of our HsGDY electrode is not the main part owing to their small amount. We will plan to perform the systematic study of electrochemical properties of oxide HsGDY after this paper. In Fig. 1g, Raman spectrum indicated the presence of defects which origin from the edge rather than the oxidation of the structure (J. Am. Chem. Soc. 2012, 134, 8646-8654). The solid-state NMR spectrum of HsGDY is different with that of graphite oxide as shown in Figure R1 (Science 2008, 321, 1815-1817). The peaks at 75.5 and 81.1 ppm are assigned as $-C\equiv C-C\equiv C-$ which was also reported by the reference (Eur. Polym. J. 2012, 48, 705-711).

We revised those discussions in our revision (page 5 line 100-104) as below:

The XPS through the peak fitting of C1s (Fig. 1f) and Fourier transform infrared spectroscopy (FT-IR) measurement (Fig. 1g) show the existence of C-O and C=O bonds, but only weak peaks could be observed in solid-state NMR around 67.5 and 166.0 ppm (Fig. 1d). Those results indicated the existence of small amount of C-O and C=O bonds on the surface of HsGDY samples. The origin of the C-O and C=O bonds might be ascribed to the chemical adsorption of oxygen on the surface of HsGDY and the reaction between oxygen and some exposed terminal acetylenic bond, which was also observed in other carbon materials^{15,25}.

Figure R1. The solid-state NMR spectra of (a) HsGDY and (b) Graphite Oxide (Science 2008, 321, 1815-1817).

3. Stating a loading density for MCOF to be 1 mg does not make sense and it must be given per volume or area.

A: Thanks for your suggestion. The areal loading density is added in Table 1.

4. Equation (1) does not show the mechanism of the lithium storage and is not useful. This should be removed. The discussion below that is more useful although it should be justified more thoroughly.

A: Thanks for your suggestion. The equation has been removed.

5. On page 13 it is stated that the definition of binding energy is given in the Supplementary information but this not the case. More details of the theoretical calculations and results are needed. The units should be given in 15(b). Is the result in 15(d) the total binding energy of all Li or per atom? How are the configurations in 15(b) and 15(d) selected and how is the storage capacity determined? Similar comments apply to 16(b).

A: Thanks for your comments. The definition of binding energy and the unit are provided in the figure captions of Supplementary Figures 15-17, as follows: $E_b = (E_{\text{HsGDY}} + nE_{\text{Li/Na}} - E_{\text{complex}})/n$, where E_{HsGDY} , $E_{\text{Li/Na}}$, E_{complex} are the energies of HsGDY (C_{24}H_6), single Li/Na atom, and $n\text{Li}(\text{Na})/\text{HsGDY}$ complex, respectively; n is the number of adsorbed Li/Na atoms. More details of theoretical calculations and results are provided in additional discussion and the Supplementary Figures 15-17 in revised supplementary information. In the calculations, the binding energy is given for per Li/Na atom. Before constructing the representative adsorption structures of Li/Na on HsGDY in Supplementary Figures 16-17b, six possible sites for the storage of single Li/Na atom are confirmed by the appropriate binding energy (Supplementary Figure 15). Furthermore, the binding energies of multiple equivalent Li/Na atoms on six different positions of HsGDY are calculated (Supplementary Figure 16-17a). The result indicates the symmetrical configuration is benefit for the stability. However, it is impossible to fill all the above possible storage sites at the same time because of the repulsion interaction between Li/Na atoms in nearest-neighbor sites. Following the

above principles, and guaranteeing the most stable binding energies and maximization of Li storage, a stable $\text{Li}_{28}\text{-C}_{24}\text{H}_6$ complex (configuration II) is selected as the optimized configuration. The binding energy for per Li atom in this configuration is calculated to be 1.39 eV, which is even more stable than that of single type of Li atoms adsorbed configuration (Supplementary Figure 15). The result demonstrates the configuration is reasonable. Furthermore, the storage capacity is calculated to be 2553 mAh g^{-1} by means of the $\text{Li}_{28}\text{-C}_{24}\text{H}_6$ configuration. The same strategy is applied in the calculation of Na storage. However, Na displays stronger repulsion and larger atomic diameter than those of Li, which lead that the number of Na which can be put in the storage sites is less than that of Li. In consequence, a stable $\text{Na}_{22}\text{-C}_{24}\text{H}_6$ (configuration III) is obtained as shown in Supplementary Figure 17.

The additional discussion is provided in main text as follows:

More details of theoretical calculations and results are provided in Supplementary Fig. 15-17 of Supplementary Information. (Page 13 line 310-311)

The additional discussion and the Supplementary Figures are provided in supplementary information as follows:

Supplementary Figure 15. The calculated binding energies of single Li/Na atom at different sites on selected repeating unit, (a) top view and (b) cross-section view of Li/Na- C_{24}H_6 complex and (c) its binding energy (E_b). Here $E_b = (E_{\text{HsGDY}} + nE_{\text{Li/Na}} - E_{\text{complex}})/n$, where E_{HsGDY} , $E_{\text{Li/Na}}$, E_{complex} are the energies of HsGDY (C_{24}H_6), single Li or Na atom, and $n\text{Li}(\text{or Na})/\text{HsGDY}$ complex, respectively; n is the number of adsorbed Li or Na atoms.

The DFT calculation method is provided in experimental section at the beginning of the supplementary information. Before constructing the representative adsorption structures of Li/Na on HsGDY in Supplementary Figures 16-17b, six possible sites for the storage of single Li/Na atom are selected and confirmed by the appropriate binding energy (Supplementary Figure 15).

Supplementary Figure 16. (a) The calculated binding energies of multiple equivalent Li atoms at one site on selected repeating unit. (b) Representative adsorption structures and corresponding binding energies (E_b) and storage capacity for Li atoms on HsGDY. $E_b = (E_{\text{HsGDY}} + nE_{\text{Li}} - E_{\text{complex}})/n$, where E_{HsGDY} , E_{Li} , E_{complex} are the energies of HsGDY (C_{24}H_6), single Li atom, and $n\text{Li}/\text{HsGDY}$ complex, respectively; n is the number of adsorbed Li atoms. Capacity = $nF/3.6M$, where $F = 96500$ mAh and M is the mass of HsGDY (C_{24}H_6) in the unit of gram.

Supplementary Figure 17. (a) The calculated binding energies of multiple equivalent Na atoms at one site on selected repeating unit. (b) Representative adsorption structures and corresponding binding energies (E_b) and storage capacity for Na atoms on HsGDY. $E_b = (E_{\text{HsGDY}} + nE_{\text{Na}} - E_{\text{complex}})/n$, where E_{HsGDY} , E_{Na} , E_{complex} are the energies of HsGDY (C_{24}H_6), single Na atom, and $n\text{Na}/\text{HsGDY}$ complex, respectively; n is the number of adsorbed Na atoms. Capacity = $nF/3.6M$, where $F=96500$ mAh and M is the mass of HsGDY (C_{24}H_6) in the unit of gram.

Furthermore, the binding energies of multiple equivalent Li/Na atoms on six different positions of HsGDY are calculated (Supplementary Figure 16-17a). The result indicates the symmetrical configuration is benefit for the stability. However, it is impossible to fill all the above possible storage sites at the same time because of the repulsion interaction between Li/Na atoms in nearest-neighbor sites. Following the above principles, and guaranteeing the most stable binding energies and maximization of Li storage, a stable $\text{Li}_{28}\text{-C}_{24}\text{H}_6$ complex (configuration II) is selected as the optimized configuration. The binding energy for per Li atom in this configuration is calculated to be 1.39 eV, which is even more stable than that of single type of Li

atoms adsorbed configuration (Supplementary Figure 15). The result demonstrates the configuration is reasonable. Furthermore, the storage capacity is calculated to be 2553 mAh g⁻¹ by means of the Li₂₈-C₂₄H₆ configuration. The same strategy is applied in the calculation of Na storage. However, Na displays stronger repulsion and larger atomic diameter than those of Li, which lead that the number of Na which can be put in the storage sites is less than that of Li. In consequence, a stable Na₂₂-C₂₄H₆ (configuration III) is obtained as shown in Supplementary Figure 17.

6. The paper still has problems associated with the English expression which make it hard to understand in places. A few examples are: on page 6, the sentence “Compared with that of triethynylbenzene monomer (Supplementary Fig. 5), Raman spectrum of HsGDY shows increasing intensity of graphic C=C stretching vibration in G-band.” and “the high degree of π -conjugated system which”; on page 8 “The loading density of the film is weighed about 0.10-0.11 mg cm⁻² when the thickness is about 700 nm” These are other issues need to be fixed.

A: Thanks for your advice. The issues mentioned by the reviewer have been corrected. We have also carefully revised some of the other wrong expressions and grammatical errors in English in this revised version.

Point by Point Responses:

Reviewer #3 (Remarks to the Author):

The authors have modified their manuscript in response to the comments of the referees.

A: Thank you very much for your very thoughtful review of our work. We really appreciate all of your comments in support of this study.